# Frontal lobe-related cognition in the context of self-disgust

**Vasileia Aristotelidou[1,2], Paul G. Overton[1]***, **Ana B. Vivas[3]**

**1** Department of Psychology, University of Sheffield, Sheffield, United Kingdom, **2** South East European Research Center, SEERC, Thessaloniki, Greece, **3** Department of Psychology, CITY College, University of York Europe Campus, Thessaloniki, Greece

* p.g.overton@sheffield.ac.uk

## Abstract

Self- disgust is an adverse self-conscious emotion that plays an important role in psychopathology and well-being. However, self-disgust has received little attention in the emotion literature, therefore our understanding of the processes underlying the experience of self-disgust is relatively scarce, although neuropsychological and neuroimaging studies support the idea that this emotion may heavily rely on frontal lobe-related cognition. To test this hypothesis, in two studies we investigated the relationship between state and trait levels of self-disgust, cognition and emotion regulation in healthy adults. Specifically, in Study 1 we tested the hypothesis that emotion regulation strategies (avoidance, suppression, and cognitive reappraisal) mediate the relationship between inhibition ability and state and trait levels of self-disgust. In Study 2, we followed a more comprehensive approach to test the hypothesis that frontal lobe-related cognitive processes (updating, Theory of Mind–ToM-, and self-attention) are closely related to the experience of self-disgust in healthy adults. Overall, across these studies, we found evidence to support the idea that inhibition ability and ToM may play a role in the experience of state and trait self-disgust, respectively. However, we did not find consistent evidence across the two studies to support the notion held in the literature that the experience of self- conscious emotions, in this case self-disgust, is heavily dependent on frontal lobe-related cognition.

## Introduction

Disgust is a basic emotion and is considered to reflect the natural, instinctual need to avoid contact with pathological microorganisms, as well as human and animal metabolic by-products [1, 2]. Despite disgust being considered to be an evolutionarily adaptive mechanism, studies propose that abnormal levels of disgust are related to various psychological disorders including depression, anxiety, contamination-based Obsessive Compulsive disorder (OCD), eating disorders, sexual dysfunctions and hypochondriasis [3–6]. In the recent years, a different disgust-related construct has been investigated, termed *self- disgust*, originally proposed by [7]. Self- disgust describes the emotion of disgust but directed towards one's own physical appearance, moral actions, and behaviour [8]. This more cognitively demanding emotion [9] belongs to the broader category of negative self- conscious emotions (SCEs), along with

**Data Availability Statement:** The data can be found here https://doi.org/10.15131/shef.data.23569269.v1.

**Funding:** South East European Research Centre (https://www.seerc.org) provided partial PhD studentship funding to Vasileia Aristotelidou The

funders had no role in study design, data collection and analysis, decision to publish, or preparation of the manuscript.

**Competing interests:** The authors have declared that no competing interests exist.

shame, guilt and embarrassment. Similar to basic disgust, altered levels of self- disgust are related to psychopathology [10–12]. Like other SCEs, such as shame, self- disgust is considered to serve a social conformity role, as it promotes socially beneficial behaviour, by regulating one's emotions, thoughts, and actions [13].

Despite its importance for well-being, self-disgust has received little attention in the emotion literature, and our understanding of the processes underlying the experience of self-disgust is relatively scarce. There is some evidence, mostly from neuropsychological and neuroimaging studies, to support the idea that SCEs depend on higher-order cognitive processes, such as executive function (EF) and emotion regulation (ER) strategies, that heavily rely on the frontal lobe [14–17]. EF is an umbrella term describing high- order cognitive processes that support goal directed behavior [18]. According to a widely accepted model [19], EF includes three independent components; *inhibition* (the ability to inhibit unwanted responses), *shifting* or *cognitive flexibility* (the ability to shift between mental states) and *updating* (the ability to update contents in working memory). However, in a latter version, Miyake and Friedman removed inhibitory control as a separate domain of EF [20], and suggested that some level of inhibition might be involved in all EF processes (see also Diamond for a similar proposal [18].

Previous research supports the idea that EF processes contribute to basic emotions such as disgust [21, 22], but research regarding SCEs is limited and focused mostly on guilt and shame. Keith et al. found that better self-reported cognitive flexibility, as measured with the *Cognitive Flexibility Scale* (CFS), was associated with and predicted lower trait levels of guilt in veterans, after controlling for Post-Traumatic Stress Disorder (PTSD) symptoms [23]. Muris et al. found that better behavioural inhibition, as measured with the *Behavioral Inhibition Questionnaire* (BIQ), was associated with higher trait levels of shame and guilt in pre-school and school children [24]. Marcusson-Clavertz et al. examined the relationship between a guilty-dysphoric daydreaming style, a maladaptive day- dreaming style referring to aggressive and guilty thoughts towards oneself, and updating skills, measured with the *Short Imaginal Processes inventory* and the *n- back* task. The authors found that better updating skills significantly predicted more frequent use of guilty-dysphoric style [25].

The only study, so far, that investigated the relationship between trait self-disgust and EF [26] found that worse EF ability, as measured by the *Verbal Fluency* and *Trail Making Test-Part B*, was correlated with higher levels of self- disgust and lower levels of guilt in patients with schizophrenia and healthy adults. One explanation as to why EF ability may be related to the experience of SCEs could be that these complex cognitive processes are needed to generate the emotion. Tracy and Robins have proposed that the generation of SCEs upon the occurrence of an event (e.g., failure in the exams) requires someone to consciously evaluate and represent the information in relation to the self, and identify the congruency/ incongruency between the elicited self-representation (e.g., *failure in the exams*) and the stable self-representation or identity-goal (e.g., *I am a successful student* or *I want to be a successful student*). Thus, inhibitory control may be needed to select and compare these two representations [27]. Another way EF ability may relate to the experience of self-disgust is via emotional regulation. We know that individual differences in EF significantly predict successful regulation of basic emotions [28], with working memory capacity being the most reliable predictor.

We hypothesise that inhibition and updating in working memory may play a role in regulation SCEs due to their relevance and close relationship to the regulation of basic emotions (e.g. [29–31]). However, the few studies conducted so far to investigate the relationship between EF and SCEs have mostly focused on clinical populations [32–34], and none of the studies have investigated both trait and state self-disgust. Here we investigated the relationship between inhibition (stop signal task, Study 1) and updating in working memory (n-back task, Study 2)

and state (narration emotion induction paradigm) and trait measures of self-disgust in healthy adults.

Successful ER is critical for psychosocial functioning and adjustment and is positively correlated with mental health [35, 36]. It is widely accepted that there are three main ER strategies; *suppression*, *re-appraisal* and *avoidance/acceptance* (see [30] for a review). *Expressive suppression* refers to active inhibition of emotional reactions including facial, verbal expressions and gestures [37, 38], whereas *cognitive reappraisal* involves the re-interpretation of emotional events. It usually occurs at relatively early stages of the emotion generation process, and is associated with the recruitment of a widely distributed frontal cortical network [39, 40]. Finally, *experiential avoidance* has been less well researched and refers to one's unwillingness to experience thoughts, emotions and physiological reactions [41], and in particular the adversely evaluated ones (e.g., fear and anxiety eliciting) [42]. Another essential distinction to be made is between the two components of ER strategies; the frequency and the efficiency [43]. The former describes the everyday use of the preferred ER strategy, whereas the latter describes how successfully the chosen ER strategy is used [44, 45], in response to a specific emotional stimulus. Negative SCEs are usually elicited through highly disturbing memories including emotional, sexual or physical abuse, often leading to psychopathology such as anxiety and depression [10–12]. These experiences are often so aversive that they are eventually incorporated into one's self- identity. Consequently, SCEs are especially susceptible to regulation [46]. The relationship between ER and SCEs has received some research attention. For instance, it has been shown that cognitive reappraisal efficiency is associated with lower levels of experimentally induced shame [47, 48]; while self- disgust has been positively correlated with suppression frequency and negatively with cognitive reappraisal frequency [49, 50]. It has also been suggested that ER may mediate the relationship between EF and social adjustment. Specifically, Fernandes suggested that acceptance/avoidance mediates the relationship between inhibition, measured with the *Stop Signal Task*, and emotional/ behavioural problems, measured with the *Strengths and Difficulties Questionnaire* (SDQ), in children and adolescents [51, 52] (see [46] for similar results in children with conduct disorder). Thus, in Study 1 we investigated whether ER strategies would mediate the potential relationship between inhibition ability, state and trait levels of self-disgust.

SCEs are a unique class of emotions which involve the concept of 'self' [53], and therefore self-evaluation process are needed to experience them. Specifically it has been proposed [27] that in order to experience SCEs, one needs to initially appraise one's own public and/ or personal self- representation (proximal cause), then shift attention towards the eliciting external stimulus (distal cause), and evaluate it in relation to the self. Thus, other frontal-lobe related cognitive processes that are proposed to play an important role in the generation of SCEs are self-awareness and self-attention. However, to our knowledge this hypothesis has not been tested empirically. Thus, in Study 2, we followed a more comprehensive approach to the investigation of cognitive processes that may contribute to the experience of self-disgust and explored if self-attention (measured by the self-prioritization task) would be associated (and predict) self-disgust trait levels, as both reflect day-to-day use of self- attention. Finally, we also investigated (Study 2) Theory of Mind (ToM) as a potential predictor of self-disgust levels. ToM refers to the ability to understand others' complex emotional and mental states, facial expressions, goals, beliefs and intentions [54]. Lagattuta and Thompson proposed that ToM and SCEs follow similar developmental trajectories, and that both SCEs and ToM require accurate comprehension of others' verbal signs and social behaviour, appraisal of social norms and awareness of social feedback [55]. Similarly, Zinck proposed that SCEs may depend heavily on ToM capacity [56]. Only two studies [57, 58] have examined the relationship between ToM ability and SCEs, without including self-disgust however. Park et al. [58]

concluded that ToM was a significant predictor of trait guilt, but not shame, in participants at ultra-high risk for psychosis and healthy controls, whereas Heerey et al. [57] concluded that worse ability to recognise SCEs in children with autism spectrum disorder was related to their poorer ToM ability.

There is a crucial differentiation between trait and state SCEs. In the context of self-disgust, trait is characterized by the enduring manifestation of self-disgust emotions in one's daily existence. Conversely, state denotes the experience of self-disgust that arises in response to particular stimuli or situations [8]. Previous studies [7] have indicated a negative association between better EF performance and trait self-disgust in individuals with schizophrenia. Additionally, other studies have suggested that trait self-disgust may be closely related to the frequency of use of ER strategies and ToM. On the other hand, ER efficiency and self-attention are likely to be associated with self-disgust state, as ER efficiency encompasses the incidental use of the chosen ER strategy, and self-attention reflects one's capacity to prioritize attention to oneself when confronted with irrelevant stimuli.

To sum up, in two studies with healthy adults we aimed at investigating the association between a wide range of frontal-lobe related cognitive processes [inhibition, updating in working memory, ToM, and self-attention), ER strategies and trait and state levels of self-disgust. We expect the findings of these studies to shed light on the specific cognitive mechanisms underlying this complex emotion. Self- disgust is crucial for social functioning, as it involves a sense of the self- regulation in relation to others [57]. This coupled with its close relationship with psychopathology (e.g., depression, see [12], makes it important to understand what factors may underlie altered levels of self- disgust.

## Study 1

In Study 1, we investigated the potential mediating effects of ER strategies (suppression, cognitive reappraisal, and avoidance/acceptance) on the relationship between inhibition and self-disgust, both self-reported and induced via a narration paradigm. We hypothesised that reappraisal, suppression, and avoidance would mediate the relationship between inhibition ability (as measured by the Stop Signal Task) and self- disgust levels. Specifically, we hypothesised that inhibition ability, as indexed by the stop signal reaction time (SSRT) in the Stop Signal task, would be a better predictor of self- disgust experience than NoGo accuracy, as the former takes into consideration the efficiency of performance on both Go and NoGo trials [59]. It was also expected that self- disgust trait and state would be influenced differently by ER strategies, as they constitute different components of self- disgust experience [8]. Also, we assume that reappraisal would be positively correlated with inhibition ability, and negatively correlated with anxiety and depression, and self- disgust. In contrast, we hypothesised that avoidance and suppression would be negatively correlated with inhibition ability, and positively correlated with anxiety and depression, and self- disgust. Finally, we also expected that better inhibition ability (SSRT) would be positively correlated with more adaptive ER strategies (i.e., cognitive reappraisal) and negatively with less adaptive ones (i.e., suppression and avoidance).

## Materials and method

### Participants

Power analysis (G* Power software; [53]) revealed that 163 participants would be sufficient to detect a significant effect ($\alpha$ = 0.05, two tailed) for a large effect size ($\eta^2$ = 0.35; power = 0.99) in mediation analysis. A hundred and ninety-five adults were recruited through social media platforms and from a university in Northern Greece. The inclusion criteria for the participants were: i) no history of psychiatric disorders or sustained brain injury; ii) no evidence of a

history of alcohol and drug abuse; iii) not taking psychoactive medication, and iv) being between 18 and 30 years old. From the initial sample of 195 participants, 31 were excluded from the analyses: 19 due to inadequate/low quality narrations (e.g., too short or did not include specific personal experiences) in our self-disgust induction, as judged by two of the authors; 9 due to 0% accuracy on NoGo trials, and one because of incomplete data. Finally, three other participants were eliminated because they were identified as outliers in their SSRT scores (less than 2% of the total number sampled; [60]. Therefore, 163 participants with a mean age of 25.5 years old (57 males and 106 females) were included in the analyses.

The study was approved by the University of Sheffield Ethics Committee, and all participants provided informed consent via the online platform *Gorilla.sc* (www.gorilla.sc). By participating in the study, they also entered a lottery to win gift vouchers for an electronics shop.

## Measures and procedure

The *Emotion Regulation Questionnaire* (ERQ, see [55, 56] for the Greek validated version) assesses the habitual use of two widely used strategies: cognitive reappraisal (e.g., "When I want to feel positive emotion, I change what I'm thinking about") and expressive suppression (e.g., "I control my emotions by not expressing them"). It consists of 10- items using a Likert scale from 1 (strongly disagree) to 7 (strongly agree). Higher scores reflect more frequent use of cognitive reappraisal and expressive suppression. The questionnaire showed good internal consistency; $\alpha$ = .86 for cognitive reappraisal, and $\alpha$ = .74 for suppression.

The *Acceptance and Action Questionnaire*–II (AAQ2; see [61, 62] for the Greek validated version) measures psychological flexibility in ER (e.g., "I worry about not being able to control my worries and feelings"). The AAQ2 consists of 7 items that assess experiential avoidance, using a Likert scale from 1 (never true) to 7 (always true). Higher total scores represent cognitive inflexibility and avoidance, whereas lower scores represent acceptance [63]. The questionnaire showed high internal consistency, $\alpha$ = .90.

The *Self- Disgust Scale* (SDS; see [12, 64] for the Greek validated version, SDS-G) measures disgust directed at the self, including disgust at one's behavior and one's "self" (e.g., "The way I behave makes me despise myself" and "I find myself repulsive"). The SDS-G consists of 18 items (6 are fillers) with a 7-point Likert scale (1: strongly agree, 7: strongly disagree), with higher scores indicating higher level of trait self-disgust. The questionnaire showed high internal consistency, $\alpha$ = .83.

The *Hospital Anxiety and Depression Scale* (HADS; see [65, 66] for the Greek validated version) is a 14-item questionnaire to measure anxiety and depressive symptoms, which is suitable for the general population [67], with higher scores indicating higher levels of anxiety and depressive symptoms. The questionnaire showed good internal consistency, $\alpha$ = .80 for anxiety, $\alpha$ = 0.77 for depression.

The *Stop Signal Task* [68] measures response inhibition. In this task, participants have to press a key (in our case J or F) to indicate the direction of an arrow that is presented at the centre of a screen. On Go trials (75%), a white arrow is presented in the centre of the screen and participants have to press a key to indicate its direction as fast as they can without making errors. On NoGo trials (25%), the white arrow changes to red (the stop signal) and participants must withhold their response. In line with previous SST protocols [69], participants first completed a practice block, and then two experimental blocks, with a total of 227 trials. The time between the presentation of the stimulus and the stop signal, Stop Signal Delay (SSD), varied across trials so that participants were not able to predict the onset of the No-Go signal [59]. Based on Verbruggen et al., the SSD varied between 400 ms and 500 ms for both practice and experimental blocks (mean SSD for practice 440 ms, mean SSD for experimental blocks was

444 ms) [59]. As recommended by Verbruggen et al., brief feedback (thumbs up for correct answers and down for incorrect), was given after every trial, in both practice and experimental blocks [52]. Also, between the blocks participants were reminded of the instructions. There are different methods to obtain an index of the participant's ability to inhibit the initiated motor response on NoGo trials using the Stop-Signal Reaction Time (SSRT) [52]. We employed the *integration method*, which is thought to be more reliable and less biased than the *mean method* [70]. In addition to the SSRT, NoGo accuracy was also employed as an index of successful inhibition.

The *Narration emotion-induction paradigm* was based on Dickerson et al.'s paradigm [71] (see also [72]). Participants were asked to write down a personal experience which elicited the feeling of self- disgust and, as a control state, to describe what they did the previous day. The instructions given for the neutral narration were as follows: "I want you to write a few sentences about what you did yesterday, for example I went shopping, went to the grocery store and visited my family". Whereas the instructions for the self-disgust narration were as follows: "I want you to write a few sentences about the most shocking and disturbing incident that you have ever experienced during your lifetime; you are kindly asked to emphasize particularly the part of the story that made you feel disgusted about yourself and or a personal experience which elicited the sense of "repulsiveness" towards yourself. The important thing is that you declare your deepest thoughts and feelings. This could be a breakup or a negative change in your body which made you feel repulsed by yourself. Ideally, whatever you speak about should deal with an event or experience that you have not talked with others about in detail".

After each narration condition, participants were asked to self-report how they felt using a Visual Analogue Scale (VAS) from 0 (not at all) to a 100 (Extremely), for the target emotion (self-disgust) and other non-target emotions (anger, happiness, and sadness). The neutral narration was always presented before the self- disgust narration. The following two measures of self-disgust were included in the analyses, VAS self-disgust narration (SD state) and VAS difference score (VAS self- disgust narration minus VAS neutral narration; SD diff) to take into account baseline differences [73].

The study was run online using Gorilla and it consisted of three parts: self-report measures —demographics, ERQ, AAQ, HADS and SDS; the Stop Signal Task; the Narration Induction Paradigm. The full study lasted approximately 30 minutes and could only be completed on a PC. Participants were also instructed to complete the study in a quiet environment.

## Results and discussion

All measures were checked for normality. Absolute z scores were calculated for kurtosis and skewness, with an absolute z score cut- off point at 3.29 [74]. All variables were found to be normally distributed, except for SD state, SD diff and SSRT. Therefore, these variables were reflected and then transformed using the standard square root method. Since our data were moderately skewed, and our data were whole number counts, square root transformation was optimal [75, 76]. All the variables were normally distributed after the transformation. To interpret the results, the variables were re- reflected using the same method.

In order to test if the emotion-induction manipulation was effective at eliciting self-disgust, we submitted the VAS self-disgust scores to a one-way repeated measures ANOVA with time (pre- and post-) as the within-subject factor. Self-disgust levels were significantly higher after the induction (Mean = 74.99, SD = 27.89), relative to baseline (Mean = 17.07, SD = 21.47), $F(1, 162) = 528.13$, $p < .001$, $\eta^2 = 0.765$.

Pearson bi-variate correlations were then conducted to test for associations between trait (SDS-G) and state self-disgust (SD state, SD diff), ER strategies (ERQ_R, ERQ_S, and AAQ),

**Table 1. Inter- correlations between self- disgust, emotion regulation frequency, inhibition ability and negative affect.**

| Variable | | 1 | 2 | 3 | 4 | 5 | 6 | 7 | 8 | 9 | 10 | 11 |
|---|---|---|---|---|---|---|---|---|---|---|---|---|
| 1. Age | r | — | | | | | | | | | | |
| 2. ERQ R | r | 0.044 | — | | | | | | | | | |
| 3. ERQ S | r | 0.003 | 0.077 | — | | | | | | | | |
| 4. Sum AAQ | r | -0.123 | -0.265** | 0.285** | — | | | | | | | |
| 5. HADS TOTAL | r | -0.147 | -0.419** | 0.24* | 0.533** | — | | | | | | |
| 6. HADS (A) | r | -0.163* | -0.374** | 0.196* | 0.559** | 0.912** | — | | | | | |
| 7. HADS (D) | r | -0.101 | -0.385** | 0.24* | 0.403** | 0.9** | 0.642** | — | | | | |
| 8. SDS—G | r | -0.164* | -0.309** | 0.257** | 0.490** | 0.697** | 0.643** | 0.62** | — | | | |
| 9. SD state | r | 0.003 | -0.047 | 0.084 | 0.023 | 0.218* | 0.141 | 0.257** | 0.267** | — | | |
| 10. SD state diff | r | 0.059 | -0.218* | 0.025 | 0.089 | 0.192* | 0.113 | 0.238* | 0.229* | 0.261** | — | |
| 11. SSRT | r | 0.006 | 0.013 | 0.103 | 0.237* | 0.141 | 0.142 | 0.113 | 0.124 | -0.164* | 0.059 | — |
| 12. NoGo accuracy | r | -0.023 | 0.016 | -0.087 | -0.234* | -0.154* | -0.155* | -0.123 | -0.099 | 0.209* | 0.015 | -0.964* |

$*p < .05$,

$**p < .001$;

ERQ S: Emotion Regulation Questionnaire Suppression; ERQ R: Emotion Regulation Questionnaire Reappraisal; HADS: Hospital Anxiety (A) and Depression (D); SDS —G: Self Disgust Self report total scores; SD state: VAS self-disgust post- induction; SD state diff: VAS self- disgust—minus VAS neutral; SSRT: Stop Signal Task Delay.

inhibition (SSRT, and NoGo accuracy), and depression and anxiety (Table 1). SSRT was positively correlated with the AAQ (r = 0.237, p = .002), and negatively with SD state (r = -0.164, p = .037). That is, participants who had worse inhibitory skills (needed more time to inhibit their responses) used the avoidance strategy more frequently to regulate their emotions and reported lower levels of state self-disgust. NoGo Accuracy was negatively correlated with the AAQ (r = -0.234, p = .003), HADS total (r = -0.154, p = .052), SSRT (r = -0.954, p< .001) and positively correlated with SD state (r = 0.209, p = .007). So, participants who were more successful in inhibiting their response used the avoidance strategy more often, had lower levels of anxiety and depression, needed more time to inhibit their responses during the SST and reported higher levels of state self- disgust. SD diff was negatively correlated with ERQ R (r = -0.218, p = .005). That is, participants who reported greater self- disgust levels in the emotion condition relative to neutral condition, used cognitive reappraisal less frequently. SDS- G was positively correlated with the AAQ (r = 0.490, p < .001), ERQ S (r = 0.257, p< .001) and HADS total (r = 0.697, p< .001) and negatively with age (r = -0.164, p = 0.037) and ERQ R (r = -0.309, p< .001). Consequently, participants who reported higher trait levels of self- disgust, predominantly used avoidance and suppression strategies to regulate their emotions, while they used cognitive reappraisal less, had higher levels of anxiety and depression, and were of a younger age.

We then conducted mediation analysis using JASP (version 0.14.1; [77] to examine our primary hypothesis with regard to the use of cognitive reappraisal, suppression and avoidance strategies mediating the relationship between inhibition (SSRT) and trait and state levels of self- disgust. It order to conduct mediation analysis three conditions should be met; i) the predictor variable (SSRT or NoGo accuracy) should significantly predict both the mediator (a pathway), in this case frequency of use of cognitive reappraisal, suppression and avoidance, and the outcome (c or direct pathway), in this case trait (SDS -G-) and state (SD state and SD diff) self-disgust; ii) the path from the mediator to the outcome should be significant, when controlling for the predictor (b or indirect pathway). Thirdly, the pathway from the predictor to the outcome should be significantly reduced when controlling for the mediator (c' or total

pathway). The described pathways were assessed by using the correlations above and Structural Equation Modeling (SEM) [78]. Moreover, to increase the confidence level we used both delta method standard errors and bias-corrected percentile bootstraping to calculate main and interaction effects (on 1000 bootstrap samples), along with their significance levels and a 95% confidence interval [79]. Only cases where all three conditions described above as being necessary preconditions for mediation were met are considered below. In cases where all three conditions were satisfied, Sobel's test was conducted to evaluate if the reduction in the predictor to outcome pathway, when controlling for the mediator (c' or total pathway), was significant, which would indicate a mediation effect [80].

In the first model (see S1 Fig and S1 Table), SSRT was used to predict SD state with frequency of use of the avoidance strategy -AAQ- as the mediator. SSRT was positively related to AAQ, a = 0.499, p = 0.002 (a pathway). The direct pathway between SSRT and SD state (c pathway) was found to be statistically significant as well, c = -0.034, p = 0.034. However, the indirect pathway between the mediator (AAQ) and SD state, controlling for SSRT (b pathway), b = 0.003, p = 0.425, was found to be non- significant. Consistent with that, the relationship between SSRT and SD state (c' pathway) was still significant after controlling for AAQ, c' = -0.034, p = 0.024. After adjusting for the effect of depression and anxiety (HADS total score), the indirect pathway between the mediator AAQ and the SD state, controlling for SSRT (b pathway), b = - 0.015, p = 0.359, was found again to be non- significant.

In the second model (see S2 Fig and S1 Table), NoGo trials accuracy was used to predict SD state, with frequency of use of the avoidance strategy–AAQ as the mediator. NoGo accuracy was negatively related to AAQ, a = -12.300, p = 0.003 (a pathway). The direct pathway between NoGo accuracy and SD state (c pathway) was found to be statistically significant as well, c = 2.923, p = 0.006. However, the indirect pathway between AAQ and SD state, controlling for NoGo accuracy variable (b pathway), b = -0.248, p = 0.356, was found to be non- significant. Consistent with that, the relationship between NoGo accuracy and SD state (c' pathway) was still significant, after controlling for AAQ, c' = 3.171, p = 0.004. After adjusting for the effect of depression and anxiety (HADS total score), the indirect pathway between the mediator AAQ and SD state, controlling for NoGo accuracy (b pathway), b = 0.174, p = 0.399, was found again to be non- significant.

The key finding was that although inhibition (SSRT and NoGo accuracy) was significantly associated with both state self-disgust and the ER strategy of avoidance, we did not find significant mediation effects between SSRT/NoGo accuracy (predictors), SD state (outcome) and avoidance (mediator). In addition, and contrary to our predictions, trait levels of self-disgust did not significantly correlate with inhibition, but it did correlate with all the ER strategies. That is, higher trait levels of self-disgust were associated with higher frequency use of suppression and avoidance, and lower frequency use of cognitive reappraisal strategies. Inhibition was found to be associated with state self-disgust, in agreement with the few neuropsychological studies that report altered state experience of embarrassment (see [73, 74] for evidence in patients with frontal damage and fronto-temporal dementia, respectively) and self-disgust (see [64] for evidence with Parkinson's disease patients) in patients with executive functions deficits. However, the lack of an association between inhibition and trait levels of self-disgust is not in agreement with the higher trait levels of self-disgust found in Parkinson's disease [64] and patients with schizophrenia [26], relative to healthy controls. One should be cautious, nevertheless, when extrapolating findings from neurological patients to healthy, typically developing, populations.

Based on our results, only the use of avoidance was significantly associated with inhibition, but this ER strategy did not mediate the relationship of inhibition with state levels of self-disgust. As discussed above, inhibitory control may be needed to select and compare the elicited

self-representation (e.g., *failure in the exams*) and the stable self-representation or identity-goal (e.g., *I am a successful student*, or *I want to be a successful student*). It should be noted though that inhibition was weakly correlated with state self-disgust in our study.

## Study 2

Given the lack of significant mediation effects in Study 1, and that inhibition ability was only significantly associated with state levels of self-disgust, in Study 2 we followed a more comprehensive approach to the investigation of cognition and self-disgust experience. Thus, we included a wide range of frontal lobe-related processes that have been suggested in the literature to play a role in the generation and/or regulation of SCEs. Specifically, we included another key component of EF, that is updating in working memory, which has been found to significantly predict successful regulation of basic emotions [28]. In addition, we investigated affective ToM and self-attention which have been proposed to play a key role in the generation of SCEs [27].

Finally, McRae showed that the relationship between the frequency of using an ER strategy and the effectiveness of that strategy in real life may not be straightforward. That is, people may more frequently use a particular strategy because they do it successfully, but on the other hand, if a particular strategy is used very successfully this may reduce the need for using it frequently [81]. Given the significant relationship found between cognitive reappraisal frequency and self-disgust levels in Study 1, and the complex relationship between frequency of use of a strategy and effectiveness, we also investigated the effectiveness of positive reappraisal to regulate experimentally induced self-disgust state levels.

Thus, in this study we wanted to test the hypothesis that state and trait levels of self-disgust in healthy adults are closely linked to higher order cognition. Based on previous studies, we expected that state and trait self-disgust levels would be positively and negatively associated with (and be predicted by) ToM and updating ability, respectively. Regarding self-attention, although theoretically it has been proposed that it plays an important role in the generation of SCEs, there are no previous studies investigating self-attention in the context of SCEs. Based on the theoretical understanding of SCEs [27], we expected that a stronger attentional bias to the self would be associated with (and predict) higher levels of self-disgust trait. As self-attention has not been previously examined in relation to SCEs, our third hypothesis posits that trait self-disgust, rather than state self-disgust, will be predicted by self-attention bias. This is based on the understanding that the self-prioritization effect and trait self-disgust reflect the salience of self-attention and self-disgust, respectively, in daily life. Lastly, evidence supports the idea that cognitive reappraisal is an effective ER strategy to regulate adverse SCEs experiences such as shame [47, 82]. Thus, we hypothesise that cognitive reappraisal efficiency will be associated with, and predict, self- disgust state rather than trait.

## Materials and methods

### Participants

Power analysis (G* Power software; [64]) revealed that 59 participants would be sufficient to detect a significant effect ($\alpha = 0.05$, two tailed) for a large effect size ($\eta2 = 0.35$; power = 0.99) in multiple regression analysis with four predictors (ToM, updating, self- attention and cognitive reappraisal). Seventy-one native Greek-speaking Cyprus nationals were recruited from educational institutions in Cyprus. The inclusion criteria for the participants were the same as Study 1. From the initial sample size, 3 participants were excluded from the analyses (consisting of less than 2% of the total number sampled): 2 because they scored below 70% accuracy in the self- prioritization task and 1 because they were an outlier in the self- prioritization task.

Therefore, 68 participants with a mean age of 23.2 years old (28 males and 40 females) were included in the study. The study was approved by the University of Sheffield Ethics Committee, and all the participants provided informed written consent. By participating in the study, they also entered a lottery to win 2 x 25 Euro gift vouchers for an electronics shop.

## Measures and procedure

The full study lasted 30–40 minutes. The participants had to complete a demographic questionnaire (specific age, gender, marital status, working and educational level), the SDS-G and the HADS [83]. The Cronbach alpha was high for the SDS-G ($\alpha = 0.84$) and for the HADS ($\alpha = 0.82$ for anxiety and $\alpha = 0.71$ for depression). In addition, participants had to complete the following:

*2-back task* [84–86] measures updating in working memory. A series of black font digits were presented on a white background screen, and participants were instructed to press 'J' on the keyboard each time the target digit matched a digit presented two trials before. Participants were presented with the instructions prior to the task. Digits were presented in pseudorandom order for 500 ms, with the same digit not presented two times in a row. Each digit was followed by a blank screen for 2,500 ms. The task consisted of a practice block, and an experimental block of 62 trials. On 20 of the 60 trials, the digit matched a digit presented two trials before.

*Reading the Mind in the Eyes* (RMET) measures the ability to correctly identify and label facial expressions from images showing the face region around the eyes. It consists of 36 images (37 including the practice image), 18 of them representing positive emotions and 18 negative ones, half of them depicting the eye area of females and half of them of males. Each image is presented with 4 possible answers, and participants are instructed to choose the most representative one to accurately describe the emotion, mental, and cognitive state of the depicted individual. In the control trials, the participants are instructed to identify the gender. Overall accuracy was taken as the total score of each participant, calculated by adding 1 point for each correct answer and 0 for the incorrect ones (maximum score 36, minimum 0) [87, 88]. Higher total scores reflected better ToM capacity [87]. In addition to overall accuracy, accuracy scores were calculated separately for positive and negative emotions.

*Self- prioritization task*. Two identical versions of the task, except for the stimuli used (which matched the gender of the participant), were employed for male and female participants [89, 90]. All faces were acquired from the Chicago faces database and were unfamiliar to participants [91]. The task consisted of a learning and experimental phase. In the learning phase, one face appeared in the centre of the screen with the labels *You*, *Friend*, and *Stranger* appearing below for 5 s and participants were instructed to indicate which label matches the face depicted. In order to familiarize themselves with the task, participants received feedback during this phase.

In experimental phase 1, each trial started with a fixation cross (400ms), followed by a face (200ms), and then a delay period of 1 s. Then, one of the three labels was presented and the participant had to indicate whether the face matched the label, pressing the keys 'F' if it did and 'J' if it did not. Each face and label were presented an equal number of times and all possible combinations were counterbalanced. Total number of trials in this phase was 90. The self-face prioritization effect (SFP) was calculated by subtracting the mean response time to the self-face condition from the mean response time to the 'Stranger' condition, divided by the sum of the 2 conditions [92].

The same *Narration emotion-induction paradigm* (based on [71]) employed in Study 1 was used, but we added a cognitive reappraisal instruction manipulation. That is, following VAS reports to the self- disgust narration condition, participants were instructed to reappraise the

aforementioned autobiographical memory, based on Krishnamoorthy et al. and McRae et al.'s instructions [47, 93]: "*Think about the aforementioned described self- disgust eliciting experience from a different perspective from the one you used earlier. Try to tell yourself something that makes you feel less negative if possible. For example, you can try imagining ways the situation could improve for the better or identifying aspects of the situation that are not be as bad as they seem.*" The participants were given 2 min to reappraise their self- disgust experience. Then participants had to complete again a VAS targeting the same emotions. For the emotion induction manipulation, we calculated the VAS Self-disgust difference score (SD diff), by subtracting VAS self- disgust after the neutral narration from VAS self- disgust after self- disgust narration. For the positive reappraisal manipulation, we calculated the VAS Reappraisal difference score (VAS RA diff), subtracting VAS self- disgust after the self- disgust narration from VAS self- disgust after cognitive reappraisal manipulation.

This study was conducted face- to- face in a quiet environment using the Gorilla.sc platform to administer both the self-report measures and the computerised tasks.

## Results and discussion

We included the following variables in the analysis: trait self-disgust (SDS-G: SDS total score), state self- disgust (SD diff), cognitive reappraisal efficiency (VAS RA diff), updating (2- back accuracy), ToM (RMET total accuracy, RMET positive accuracy, and RMET negative accuracy), self- attention effect (SFP) and negative affect measured with the HADS Total score (HADS). All measures were normally distributed.

To test if the narration induction was effective at eliciting self-disgust, a *t*-test was conducted with VAS self-disgust scores to compare the neutral and the self-disgust narration conditions. Results showed that self-disgust levels were significantly higher after the self- disgust narration (Mean = 85.97, SD = 17.47), relative to the neutral one (Mean = 10.39, SD = 14.86), t (67) = - 31.35, p< .001, Cohen d effect size = -3.80. Similarly, to test if the reappraisal instruction was effective for down regulating self-disgust, a *t*-test was conducted with VAS self-disgust scores for the self-disgust narration to compare pre- and post- cognitive reappraisal instruction conditions. Results showed that self-disgust levels were significantly lower after the cognitive reappraisal instruction (Mean = 47.98, SD = 28.16), relative to before the instruction (Mean = 85.97, SD = 17.47), t(67) = - 12.20, p< .001, Cohen's d effect size = 1.48.

To examine our primary hypothesis about whether updating ability, ToM, self-attention, and cognitive reappraisal efficiency were associated (and predicted) narration-induced state and trait levels of self- disgust, Pearson bi-variate correlations were conducted (Table 2). All analyses were performed using JASP (version 0.14.1; JASP Team, University of Amsterdam, The Netherlands). SDS- G was positively correlated with HADS total (r = 0.355, p = 0.003) and negatively with RMET negative accuracy (r = -0.246, p = 0.043). That is, participants who reported higher trait levels of self- disgust, had higher overall negative affect and worse ToM ability for negative emotions. SD diff scores were only positively correlated with VAS RA diff (r = 0.300, p = 0.013). In other words, participants who reported higher levels of self- disgust during the induction showed a greater down-regulation (reduction) of self-disgust when using positive reappraisal. In addition, 2- back accuracy was negatively correlated with HADS total (r = -0.287, p = 0.018), positively with VAS RA diff (r = 0.318, p = 0.008) and RMET negative accuracy (r = 0.236, p = 0.053), so participants who had better updating ability reported lower overall negative affect, had better ToM ability for negative emotions and were more efficient in down-regulating self-disgust with cognitive reappraisal. RMET accuracy, overall and for negative emotions, was also positively correlated with SFP self- other (r = 0.282, p = 0.020 and r = 0.290, p = 0.016, respectively). That is, participants who had a greater attentional bias to the

**Table 2. Inter- correlations between self- disgust, ToM, updating ability, self- prioritization, and cognitive reappraisal efficiency.**

| Variable | | 1 | 2 | 3 | 4 | 5 | 6 | 7 | 8 | 9 |
|---|---|---|---|---|---|---|---|---|---|---|
| 1. Age | r | — | | | | | | | | |
| 2. HADS TOTAL | r | -0.013 | — | | | | | | | |
| 3. SDS–G | r | -0.004 | 0.355* | — | | | | | | |
| 4. VAS SD diff | r | 0.167 | 0.045 | -0.070 | — | | | | | |
| 5. VAS RA diff | r | -0.038 | -0.231* | -0.142 | 0.300* | — | | | | |
| 6. 2- back accuracy (%) | r | 0.150 | -0.287* | -0.061 | 0.008 | 0.318* | — | | | |
| 7. RMET_ToM_accuracy | r | 0.057 | -0.075 | -0.187 | 0.079 | 0.022 | 0.197 | — | | |
| 8. RMET_positive_accuracy | r | 0.130 | -0.042 | -0.012 | 0.088 | -0.061 | 0.047 | 0.705** | — | |
| 9. RMET_negative_accuracy | r | -0.012 | -0.073 | -0.246* | 0.047 | 0.071 | 0.236* | 0.873** | 0.270* | — |
| 10. SFP self- other | r | 0.059 | 0.011 | 0.013 | 0.042 | 0.174 | 0.209+ | 0.282* | 0.134 | 0.290* |

0.9< p+ < 0.5, p* < .05, p** < .001; 2- back accuracy: 2- back task correct trials; RMET total accuracy: Reading the Mind in the eyes total accuracy; RMET positive accuracy: Reading the Mind in the eyes positive emotions accuracy; RMET negative accuracy: Reading the Mind in the eyes negative emotions accuracy; SFP self- other: self- prioritization effect between self and other; HADS: Hospital Anxiety (A) and Depression (D); SDS—G: Self Disgust Self report total scores (self- disgust trait); VAS SD diff: VAS self- disgust—minus VAS neutral (self- disgust state); VAS RA diff: VAS self- disgust—minus VAS cognitive reappraisal).

self also had better affective ToM ability. Finally, VAS RA diff was negatively correlated with HADS total (r = -0.231, p = 0.058), so participants who regulated self- disgust experience more efficiently using cognitive reappraisal, also reported lower levels of negative affect. Given that our key outcome variables, state (SD diff) and trait (SDS-G) were only correlated with one of the other variables (VAS RA diff and RMET negative accuracy, respectively) we did not proceed to conduct regression models as initially planned.

Contrary to our hypothesis, and the literature that suggest that SCEs are heavily dependent on cognition, we found only a couple of significant correlations with our self- disgust measures. In line with previous studies [57, 58, 94, 95] higher levels of trait self-disgust were associated with poorer ToM ability for negative emotions. In addition, higher levels of narration-induced state self-disgust were associated with a greater reduction after cognitive reappraisal. This finding is in agreement with empirical evidence showing that efficient ER occurs in the context of particularly distressing SCEs related memories such as physical, sexual or emotion abuse [47]. Additionally, as expected, cognitive reappraisal efficiency was positively associated with updating, replicating the well- documented relationship between EF and ER strategies [96–98].

## General discussion

Most authors support the idea that unlike basic emotions, SCEs are complex emotions that heavily depend on frontal lobe-related cognition [14–17, 27]. This notion has received empirical support mainly from neuropsychological and neuroimaging studies [26, 32, 33] and from a limited number of studies that report significant associations between SCEs and EF, ToM, and ER mostly in children, and clinical populations. Self-disgust, a SCE that appears to play an important role in well-being, however, has received little attention in the emotion literature. Thus, in two studies we aimed at investigating the relations between frontal lobe-related cognition (inhibition, updating in working memory, ToM, and self-attention), ER (strategies and efficiency) and state and trait levels of self-disgust in healthy adults. Specifically, in Study 1 we tested the hypothesis that ER strategies (avoidance, suppression, and cognitive reappraisal) mediate the relationship between inhibition ability and state and trait levels of self-disgust. In Study 2, we

followed a more comprehensive approach to test the hypotheses that a range of additional frontal lobe related cognitive processes (updating, ToM and self-attention) are closely related to the experience of self-disgust in healthy adults. We also wanted to test whether positive reappraisal could be used efficiently to down-regulate the experience of self-disgust.

Overall, and contrary to our expectations and to what is supported in the literature, our findings do not support the idea that the experience of self-disgust is closely linked to frontal cortex-related cognitive function in healthy adults. Nevertheless, we did find two significant correlations with cognition. Specifically, we found that inhibition ability was significantly correlated with state levels of self-disgust (Study 1) and ToM ability for negative emotions was significantly correlated with trait levels of self-disgust (Study 2). The finding of an association between higher levels of trait self-disgust and worse ability to decode negative emotions is in agreement with two previous studies that investigated the relationship between ToM, guilt and shame [26, 47, 58, 99–101]. Thus, our study supports the idea that the ability to decode and understand emotions in others is related to the experience of trait self-disgust but not state self-disgust. ToM and SCEs require the accurate interpretation of social behavior, mental states, and emotions of others [54, 102]. The presence of a significant association between ToM for negative emotions and levels of self-disgust supports our hypothesis and aligns with limited existing evidence [57, 58].

Regarding inhibition, our study suggests that any role of inhibition in the experience of experimentally induced state self-disgust is not exerted via association with ER strategies (lack of mediation effects in Study 1) as suggested by evidence with basic emotions. Studies with basic emotions support the idea that EF, in particular inhibition, plays a key role in the use of ER strategies. For instance, Domingo and Armentia found that cognitive reappraisal significantly mediated the association between executive dysfunctions and psychological distress in students [103]. In agreement, Fatima and Shahid found that cognitive reappraisal, but not suppression, mediated the relationship between EF, especially inhibition, and Machiavellianism [104]. Wante et al. assessed the frequency of adaptive (e.g., acceptance) and maladaptive (e.g., rumination) ER strategies in relation to EF performance in adolescence [105]. The authors reported that adolescents who used mainly adaptive ER strategies also scored better in EF. In addition, maladaptive and adaptive ER strategies together mediated the relationship between EF impairment and self- reported depressive symptoms. In our study, only the use of avoidance was significantly associated with inhibition, but this ER strategy did not mediate the relationship of inhibition with state levels of self-disgust. Thus, EF, in this case inhibitory control, may be needed in other processes suggested to lead to the experience of SCEs [106–110]. Tracy and Robins have proposed that the generation of SCEs upon the occurrence of an event requires the identification of congruency/ incongruency between the elicited self-representation and the stable self-representation or identity-goal. Thus, inhibitory control may be needed to select and compare these two representations [27].

The disagreement between our findings and previous findings suggests that associations between frontal lobe-related functions and self- disgust, via ER, in psychiatric populations (especially in those with emotion dysregulation) may be shaped differently than in healthy participants. The existing literature on the connection between EF and SCEs has primarily focused on individuals diagnosed with neurological disorders. Therefore, more research is needed to understand the association between SCEs and EF and other frontal-lobe related processes in healthy participants. Therefore, more research is needed to understand the association between SCEs and EF and other frontal-lobe related processes in healthy participants.

Some studies have found contradictory results regarding the relationship between inhibition and ER. In agreement with Schmeichel et al., we propose that the most appropriate conclusion is that the strength of the relationship between EF, in this case inhibition, and

ER greatly depends on the specific type of ER experimental paradigm and EF measures used [31]. Indeed, there are inconsistent results between research protocols using diverse measures of the same EF constructs. For example, to measure inhibition ability, the Stroop task and SST are the commonest methods used. Yet, current literature proposes that Stroop tasks and SST measure different aspects of inhibition e.g., inhibition of recently learned processes in the SST compared to inhibition well-entrenched processes in the Stroop task (see [111, 112]).

The dissociation between state and trait measures of emotions in terms of underlying processes is not surprising, given that these measures are thought to reflect different aspects of emotions [8, 113, 114]. While trait emotions are generally believed to reflect individual and relatively stable tendencies to react in a certain way to similar situations [115], state emotions are thought to be momentary and strongly influenced by situational variables [116]. Trait and state measures of self-disgust were also differentially associated with ER strategies; state levels of self-disgust were significantly associated only with cognitive reappraisal, whereas trait levels of self-disgust were significantly associated with the three strategies (positively associated suppression and avoidance and negatively with reappraisal). Although ER strategies have been relatively well studied in relation to basic emotions (see [15] for a review), only a few studies have investigated them in relation to SCEs. Our findings with the trait measure of self-disgust are in agreement with previous studies [49, 50, 117] that found that the predominant use of more maladaptive regulation strategies (suppression and avoidance) is associated with higher levels of negative SCEs, whereas the more frequent use of more adaptive strategies such as cognitive reappraisal is associated with lower levels of negative SCEs. To our knowledge, no previous study has investigated state self-disgust experience and cognitive reappraisal. Our results from both studies are in agreement with previous research with basic disgust [118, 119] and support the idea that cognitive reappraisal may be a preferable (Study 1) and efficient strategy (Study 2) to down-regulate highly aversive emotions such as self-disgust.

Our studies have several limitations that need to be acknowledged. Firstly, the use of self-report measures for the frequency of use of ER strategies, self-disgust trait and state introduce potential biases, such as memory biases and response biases [120, 121]. Secondly, Study 1, was conducted online and so we had no control of conditions that could have influenced the results of this study. We aimed to minimize some of these limitations by systematically adjusting for the influence of overall negative affect, using pre- and post-experimental measures, and maintaining high Cronbach's alpha reliability across questionnaires [122].

Although previous research [30, 123] with basic emotions suggests that working memory is essential for successful regulation of negative basic emotions such as fear, we did not find a significant correlation between updating in working memory and self-disgust levels (state and trait). We also did not find a significant correlation between self-disgust levels (trait and state) and self-attention (self-prioritization bias), although self-attention has been included in models of SCEs [9, 53]. However, in line with previous research [124, 125] ToM ability was significantly correlated with both updating ability and self-attention. That is, participants with better updating ability and greater attention bias to the self were better at decoding negative emotions from the eyes. Given the limited evidence, future studies could incorporate other experimental measurements of EF, such as verbal fluency and shifting (cognitive flexibility), to investigate these two processes in relation to the experience of self-disgust. Additionally, different self-attention measures such as mirror and audience presence-induced self-attention may contribute to the experience of self-disgust.

## Conclusions

The present study is the first to investigate in a comprehensive manner the relationships between state (narration induced) and trait levels of self-disgust, cognition and ER, using a wide range of self-report and experimental task-based measures, in two relatively large samples of healthy adults. We found evidence that inhibition ability and ToM may play a role in the experience of state and trait self-disgust, respectively. However, overall, we did not find consistent evidence across the two studies to support that the experience of SCEs, in this case self-disgust, is heavily dependent on cognition. Regarding ER, our findings suggest that cognitive reappraisal is a preferable and efficient strategy to down-regulate self-disgust. This is an important finding since substantial research supports a strong association between high levels of self-disgust and psychopathology [10–12]. Our findings seem to agree with the notion that self-disgust may be an unique emotion associated with a distinct neural network from those associated with other negative SCEs such as shame and guilt [126–132]. Future research, using experimental methods, is needed to understand how, and if, high order cognition contributes differentially to the experience of self-disgust.

## Supporting information

**S1 Table. Direct and indirect effects of SSRT and NoGo accuracy on SD state with frequency of use of avoidance as a mediator.**
(DOCX)

**S1 Fig. Mediation analysis using frequency of use of avoidance as a mediator between SSRT and SD state.** Mediation analysis investigating the role of the frequency of use of avoidance (scores on the Acceptance and Action Questionnaire; AAQ) as a mediator between inhibition (Stop Signal Reaction Time; SSRT) ability and self- disgust state (SD state). Alpha pathway (a pathway) represents the effect of the predictor variable (SSRT) to mediator (AAQ), beta pathway (b pathway) represents the effect of the mediator (AAQ) to the outcome (SD state), when controlling for the predictor (SSRT) and the c prime pathway (c' pathway) represents the effect of the predictor (SSRT) to the outcome (SD state), when controlling for the mediator (AAQ).
(TIF)

**S2 Fig. Mediation analysis using frequency of use of avoidance as a mediator between NoGo accuracy and SD state.** Alpha pathway (a pathway) represents the effect of the predictor variable (NoGo accuracy) to mediator (AAQ), beta pathway (b pathway) represents the effect of the mediator (AAQ) to the outcome (SD state), when controlling for the predictor (NoGo accuracy) and the c prime pathway (c' pathway) represents the effect of the predictor (NoGo accuracy) to the outcome (SD state), when controlling for the mediator (AAQ).
(TIF)

## Author Contributions

**Conceptualization:** Vasileia Aristotelidou, Paul G. Overton, Ana B. Vivas.

**Data curation:** Vasileia Aristotelidou.

**Formal analysis:** Vasileia Aristotelidou.

**Funding acquisition:** Paul G. Overton, Ana B. Vivas.

**Investigation:** Vasileia Aristotelidou.

**Methodology:** Vasileia Aristotelidou, Paul G. Overton, Ana B. Vivas.

**Project administration:** Vasileia Aristotelidou.

**Resources:** Vasileia Aristotelidou.

**Supervision:** Paul G. Overton, Ana B. Vivas.

**Validation:** Vasileia Aristotelidou.

**Visualization:** Vasileia Aristotelidou.

**Writing – original draft:** Vasileia Aristotelidou.

**Writing – review & editing:** Paul G. Overton, Ana B. Vivas.

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
