## [Decision Letter · Decision Letter 0]

15 May 2023

PONE-D-23-07624Frontal lobe-related cognition in the context of self-disgustPLOS ONE

Dear Dr. Overton,

Thank you for submitting your manuscript to PLOS ONE. After careful consideration, we feel that it has merit but does not fully meet PLOS ONE’s publication criteria as it currently stands. Therefore, we invite you to submit a revised version of the manuscript that addresses the points raised during the review process.

We look forward to receiving your revised manuscript.

Kind regards,

Alexandra Kavushansky, PhD

Academic Editor

PLOS ONE

Journal Requirements:

Reviewers' comments:

Reviewer's Responses to Questions

**Comments to the Author**

1. Is the manuscript technically sound, and do the data support the conclusions?

Reviewer #1: Yes

2. Has the statistical analysis been performed appropriately and rigorously? 

Reviewer #1: Yes

3. Have the authors made all data underlying the findings in their manuscript fully available?

Reviewer #1: Yes

4. Is the manuscript presented in an intelligible fashion and written in standard English?

Reviewer #1: Yes

5. Review Comments to the Author

Reviewer #1: Thank you for the opportunity to review the paper on the association between self-disgust with cognition and emotion regulation strategies. The main research hypotheses were not confirmed, however, the presented studies are novel and provide a valuable contribution to our understanding of the processes underlying the experience of self-disgust. Nonetheless, the manuscript needs some adjustments.

Introduction

1. References 19 & 20 refer to the same item

2. In line 74 the Authors stated that one study investigated the relationship between SDS and EF. In line 89 the Authors wrote that none of the studies investigating the association between SCEs and EF have investigated self-disgust.

3. Lines 92 & 120 – what did the Authors mean by „experimentally induced trait level of disgust”? Wasn’t trait SDS measured before the experimental part of the research?

5. The Authors listed 3 basic EF components (and added that inhibition might be involved in all EF processes). However, the Authors decided to examine inhibition and updating, but not cognitive flexibility. Is it possible to explain why did the Authors decide to explore the aforementioned 2 components, but not cognitive flexibility? It has been found that SDS may be especially important for eating disorders or OCD, which are characterized by deficits in cognitive flexibility (see, e.g., Tchanturia et al., 2011; Gruner et al., 2017). This could suggest the need to examine also the relationship between cognitive flexibility and state/trait SDS.

6. Is it possible for the Authors to elaborate more on the differences between state and trait SDS in the context of how it might be associated with EF, ER, ToM, and self-attention? What is missing is a 1-2 sentence explanation of why both trait and state SDS were examined in relation to the abovementioned variables.

Results

Study 1

1. The Authors mentioned that in mediation analysis two SDS state indicators were aimed to be included – VAS SD state and VAS SD diff (finally only VAS SD state could be included). However, in Table 1 correlation coefficients between VS SD state and other variables were not included.

2. Some discrepancies between the results in the text and the results presented in the Figures were found. For instance for Model 1 in the text a = 0.449 and in Figure 1 it’s 0.499. In Model 2 in the text c=2.923 and in Figure 2 it’s 3.171; a = -12.300 in the text, in Figure 2 it’s 12.300. The entire manuscript, with particular attention to statistical reporting, should be thoroughly reviewed again for typographical errors and consistency in the number of decimals reported

3. I am not fully convinced that devoting so much space, including two figures presenting insignificant results, to describing insignificant mediation models is necessary. Can you justify why it is important to present them in this form?

Study 2

Measures and procedure

1. The authors could add information on how long did the study take (procedure section)

Discussion

1. In line 566, the authors stated that 'the ability to decode and understand emotions in others is related to the experience of trait self-disgust but not state self-disgust.' Could the authors expand on this conclusion by providing an explanation for why this might be the case?

2. Is it possible that the lack of significant association between inhibition and SDS state via ER strategies in this particular study (which is not in line with the results of other research on such associations with SCEs) might be associated with the design of this particular study? Could the Authors address this in the discussion?

3. In study 1 some medium to large associations were found between HADS with SDS and ER strategies. In the literature, solid foundations for linking SDS with psychological difficulties can be found (see Clarke et al., 2018). Is it possible that the associations between frontal lobe-related functions and SDS (via ER) in psychiatric populations (especially in those with significant emotion dysregulation) may be shaped differently than in healthy participants?

4. Limitations – I would also add that study 1, which included an examination of EF, was not conducted under standardized conditions. This means that uncontrolled independent variables could have influenced the results of this study.

References

Gruner, P., & Pittenger, C. (2017). Cognitive inflexibility in obsessive-compulsive disorder. Neuroscience, 345, 243-255.

Tchanturia, K., Harrison, A., Davies, H., Roberts, M., Oldershaw, A., Nakazato, M., ... & Treasure, J. (2011). Cognitive flexibility and clinical severity in eating disorders. Plos one, 6(6), e20462.

Clarke, A., Simpson, J., & Varese, F. (2019). A systematic review of the clinical utility of the concept of self‐disgust. Clinical psychology & psychotherapy, 26(1), 110-134.

6. PLOS authors have the option to publish the peer review history of their article (what does this mean?). If published, this will include your full peer review and any attached files.

Reviewer #1: No

---

## [Author Response · Author response to Decision Letter 0]

6 Jul 2023

Comment: References 19 & 20 refer to the same item.

Response: Reference 20 has been removed.

Comment: In line 74 the Authors stated that one study investigated the relationship between SDS and EF. In line 89 the Authors wrote that none of the studies investigating the association between SCEs and EF have investigated self-disgust.

Response: Thank you for spotting this. We have edited the text to make clear that no study has investigated the relationship with both state and trait measures of self-disgust. The revised text is as follows: “The only study, so far, that investigated the relationship between trait self-disgust and EF” (line 77), and “none of the studies have investigated both trait and state self-disgust” (line 96). 

Comment: Lines 92 & 120 – what did the Authors mean by „experimentally induced trait level of disgust”? Wasn’t trait SDS measured before the experimental part of the research?

Response: Thank you for spotting this - we made an error in our description, as we did not intend to refer to experimentally induced “trait” but rather to a “state”. We have edited the text as follows: “…and updating in working memory (n-back task, Study 2) and state (narration emotion induction paradigm) and trait measures of self-disgust in healthy adults” (lines 97-99). 

Comment: The Authors listed 3 basic EF components (and added that inhibition might be involved in all EF processes). However, the Authors decided to examine inhibition and updating, but not cognitive flexibility. Is it possible to explain why did the Authors decide to explore the aforementioned 2 components, but not cognitive flexibility? It has been found that SDS may be especially important for eating disorders or OCD, which are characterized by deficits in cognitive flexibility (see, e.g., Tchanturia et al., 2011; Gruner et al., 2017). This could suggest the need to examine also the relationship between cognitive flexibility and state/trait SDS.

Response: A rationale for the EF measures included in the studies has been now added to the Introduction. We made the strategic decision to focus on inhibition and updating as they have been previously implicated in the processing of negative basic emotions (e.g., 30-32). The only study investigating cognitive flexibility in relation to SCEs utilized a self-report measure rather than a behavioral one (24), so we did not include this component of EF in the present study. Therefore, we added the following: “We hypothesise that inhibition and updating in working memory may play a role in regulation SCEs due to their relevance and close relationship to the regulation of basic emotions (e.g., 30–32).” (Lines 92-94). We have also included a sentence in the Discussion regarding the need for further research looking at cognitive flexibility: “Given the limited evidence, future studies could incorporate other experimental measurements of EF, such as verbal fluency and shifting (cognitive flexibility), to investigate these two processes in relation to the experience of self-disgust.” (Lines 650-653).

Comment: Is it possible for the Authors to elaborate more on the differences between state and trait SDS in the context of how it might be associated with EF, ER, ToM, and self-attention? What is missing is a 1-2 sentence explanation of why both trait and state SDS were examined in relation to the abovementioned variables.

Response: For a clearer theoretical background and to explain of why both trait and state self- disgust were examined in relation to the abovementioned variables, the following has been added: “There is a crucial differentiation between trait and state SCEs. In the context of self-disgust, trait is characterized by the enduring manifestation of self-disgust emotions in one's daily existence. Conversely, state denotes the experience of self-disgust that arises in response to particular stimuli or situations (8). Previous studies (7) have indicated a negative association between better EF performance and trait self-disgust in individuals with schizophrenia. Additionally, other studies have suggested that trait self-disgust may be closely related to the frequency of use of ER strategies and ToM. On the other hand, ER efficiency and self-attention are likely to be associated with self-disgust state, as ER efficiency encompasses the incidental use of the chosen ER strategy, and self-attention reflects one's capacity to prioritize attention to oneself when confronted with irrelevant stimuli.” (Lines 152-162).

Comment: Results - Study 1 - The Authors mentioned that in mediation analysis two SDS state indicators were aimed to be included – VAS SD state and VAS SD diff (finally only VAS SD state could be included). However, in Table 1 correlation coefficients between VS SD state and other variables were not included.

Response: Thanks for pointing out to this omission, they are now reported. 

Comment: Some discrepancies between the results in the text and the results presented in the Figures were found. For instance for Model 1 in the text a = 0.449 and in Figure 1 it’s 0.499. In Model 2 in the text c=2.923 and in Figure 2 it’s 3.171; a = -12.300 in the text, in Figure 2 it’s 12.300. The entire manuscript, with particular attention to statistical reporting, should be thoroughly reviewed again for typographical errors and consistency in the number of decimals reported.

Response: Thank you. Text and figures have been now revised and errors/omissions corrected. For clarity purposes, the "c" pathway in both mediation models is not depicted in the supplementary figures; instead, it is described in the accompanying caption.

Comment: I am not fully convinced that devoting so much space, including two figures presenting insignificant results, to describing insignificant mediation models is necessary. Can you justify why it is important to present them in this form?

Response: Thank you for your suggestion. The two figures are now added to the supplementary material. 

Comment - Study 2 - Measures and procedure. 1. The authors could add information on how long did the study take (procedure section)

Response: The following has been added: “The full study lasted 30 - 40 minutes.” (Line 433).

Comment: Discussion - In line 566, the authors stated that 'the ability to decode and understand emotions in others is related to the experience of trait self-disgust but not state self-disgust.' Could the authors expand on this conclusion by providing an explanation for why this might be the case?

Response: Please see the response concerning state and trait above. 

Comment: Is it possible that the lack of significant association between inhibition and SDS state via ER strategies in this particular study (which is not in line with the results of other research on such associations with SCEs) might be associated with the design of this particular study? Could the Authors address this in the discussion?

Response: The possibility that our findings may be attributed to study design is an important point. To address this issue, the following was added: “Some studies have found contradictory results regarding the relationship between inhibition and ER. In agreement with Schmeichel et al., we propose that the most appropriate conclusion is that the strength of the relationship between EF, in this case inhibition, and ER greatly depends on the specific type of ER experimental paradigm and EF measures used (32). Indeed, there are inconsistent results between research protocols using diverse measures of the same EF constructs. For example, to measure inhibition ability, the Stroop task and SST are the commonest methods used. Yet, current literature proposes that Stroop tasks and SST measure different aspects of inhibition e.g., inhibition of recently learned processes in the SST compared to inhibition well-entrenched processes in the Stroop task (see 120).” (Line 605-613)

Comment: In study 1 some medium to large associations were found between HADS with SDS and ER strategies. In the literature, solid foundations for linking SDS with psychological difficulties can be found (see Clarke et al., 2018). Is it possible that the associations between frontal lobe-related functions and SDS (via ER) in psychiatric populations (especially in those with significant emotion dysregulation) may be shaped differently than in healthy participants?

Response: Again, this is an important point and we have added the following: “The disagreement between our findings and previous findings suggests that associations between frontal lobe-related functions and self- disgust, via ER, in psychiatric populations (especially in those with emotion dysregulation) may be shaped differently than in healthy participants. The existing literature on the connection between EF and SCEs has primarily focused on individuals diagnosed with neurological disorders. Therefore, more research is needed to understand the association between SCEs and EF and other frontal-lobe related processes in healthy participants.” (Lines 596-602).

Comment: Limitations – I would also add that study 1, which included an examination of EF, was not conducted under standardized conditions. This means that uncontrolled independent variables could have influenced the results of this study.

Response: Thank you. We have added the following to our limitations: “Our studies have several limitations that need to be acknowledged. Firstly, the use of self-report measures for the frequency of use of ER strategies, self-disgust trait and state introduces potential biases, such as memory biases and response biases (83,84). Secondly, Study 1, was conducted online and so we had no control of conditions that could have influenced the results of this study. We aimed to minimize some of these limitations by systematically adjusting for the influence of overall negative affect, using pre- and post-experimental measures, and maintaining high Cronbach's alpha reliability across questionnaires (85).” (Lines 634-641).

---

## [Editor Report · Decision Letter 1]

31 Jul 2023

Frontal lobe-related cognition in the context of self-disgust

PONE-D-23-07624R1

Dear Dr. Overton,

We’re pleased to inform you that your manuscript has been judged scientifically suitable for publication and will be formally accepted for publication once it meets all outstanding technical requirements.

Kind regards,

Alexandra Kavushansky, PhD

Academic Editor

PLOS ONE
---

## [Editor Report · Acceptance letter]

3 Aug 2023

PONE-D-23-07624R1 

Frontal lobe-related cognition in the context of self-disgust 

Dear Dr. Overton:

I'm pleased to inform you that your manuscript has been deemed suitable for publication in PLOS ONE. Congratulations! Your manuscript is now with our production department. 

Kind regards, 

on behalf of

Dr. Alexandra Kavushansky 

Academic Editor

PLOS ONE